# A Guide for the Food Industry to Meet the Future Skills Requirements Emerging with Industry 4.0

**DOI:** 10.3390/foods9040492

**Published:** 2020-04-14

**Authors:** Tugce Akyazi, Aitor Goti, Aitor Oyarbide, Elisabete Alberdi, Felix Bayon

**Affiliations:** 1Department of Mechanics, Design and Organisation, University of Deusto, 48007 Bilbao, Spain; aitor.oyarbide@deusto.es; 2Deusto Digital Industry Chair, Department of Mechanics, Design and Organisation, University of Deusto, 48007 Bilbao, Spain; aitor.goti@deusto.es; 3Department of Applied Mathematics, University of the Basque Country UPV/EHU, 48013 Bilbao, Spain; elisabete.alberdi@ehu.eus; 4Sidenor Aceros Especiales, SLU, 48970 Bilbao, Spain; felix.bayon@sidenor.com

**Keywords:** skills, workforce, food industry, Industry 4.0, digitalisation

## Abstract

The food industry has recently faced rapid and constant changes due to the current industrial revolution, Industry 4.0, which has also profoundly altered the dynamics of the industry overall. Due to the emerging digitalisation, manufacturing models are changing through the use of smart technologies, such as robotics, Artificial Intelligence (AI), Internet of Things (IoT), machine learning, etc. They are experiencing a new phase of automation that enables innovative and more efficient processes, products and services. The introduction of these novel business models demands new professional skills requirements in the workforce of the food industry. In this work, we introduce an industry-driven proactive strategy to achieve a successful digital transformation in the food sector. For that purpose, we focus on defining the current and near-future key skills and competencies demanded by each of the professional profiles related to the food industry. To achieve this, we generated an automated database of current and future professions and competencies and skills. This database can be used as a fundamental roadmap guiding the sector through future changes caused by Industry 4.0. The interest shown by the local sectorial cluster and related entities reinforce the idea. This research will be a key tool for both academics and policy-makers to provide well-developed and better-oriented continuous training programs in order to reduce the skill mismatch between the workforce and the jobs.

## 1. Introduction

The term “food industry” encompasses any company that produces, processes, manufactures, sells and serves food, beverage and dietary supplements [1]. It refers to all stages of the process, including design, construction, maintenance and delivery of solutions to the customer in the industry of animal nutrition and the food industry (food and drink) [2].

In recent times, the food industry has gone through rapid and constant changes caused by the recent industrial revolution, Industry 4.0. The term “Industry 4.0” has been used to refer to the innovative production processes which are partly or completely automated via technology, and devices communicating autonomously with each other along the value chain activities [3]. Thus, it is basically based on intelligent networking of machines, electrical equipment and novel Information Technology (IT) systems enabling processes optimisation and increased productivity of value creation chains [4,5]. The digital transformation is the key element of the ongoing industrial revolution [4,6]. Here, the digitalisation concept does not refer to a simple transfer from “analogic” to digital data and documents. It rather represents the networking between the created interfaces, the business processes, the data exchange and management [7]. Therefore, rapidly growing digitalisation has been extremely transforming the dynamics of most industries, including the food industry. The manufacturing models are changing through the development of smart technologies such as advanced robotics, a new generation of sensors, Artificial Intelligence (AI), Big Data, Internet of Things (IoT), Machine Learning, Cloud Computing, Machine to Machine (M2M) communication etc. The use of these key enabling technologies (KETs) facilitates a new phase of automation that results in innovative and more efficient processes, products and services. The aforementioned digital technologies could be ex novo applied to a new plant, as well as can be adapted to existing plants [8]. 

Up to now, despite all the challenges, Industry 4.0 has been regarded as a great opportunity for the progress of the food sector [2,9,10,11]. There have been successful attempts to keep up with the next industrial revolution. Technological developments have enabled higher efficiency rates during the manufacturing stage and lower production costs, thus, generating products with greater added value, which is critically important when there is a high level of competition among manufacturers [9]. Moreover, food security has recently been a great concern and food safety has been a top priority internationally. IoT technology, one of the key technologies of Industry 4.0, has been proven to be a solution to this concern, as it allows to identify the product and provides its traceability from cultivation to the production chain during food processing [10]. It was also demonstrated that the adoption of Industry 4.0 within the food supply chain environment of the food sector could contribute immensely to the achievement of sustainability [11]. The integration of more 4.0 technologies such as AI, Big Data, Machine Learning, M2M (and others mentioned in the previous paragraph) will lead to a faster industrial transformation in the food sector. It will change the business abruptly, facilitating the production of higher quality food products in a shorter time and at a lower cost.

The main condition for defining the expected evolution of skills requirements is to draw a general portrait of the future food industry by clarifying the industrial changes brought up by Industry 4.0. First of all, thanks to the real-time data provided by the smart and automated production systems, the workers will be able to make more accurate decisions in a short time, dealing with complex situations in the near future. Due to the advanced robotics technology, simple and monotonous tasks will be taken by collaborative robotic systems while operators carry out more qualified work and make critical decisions [12]. Concurrently, the significance of human intervention in the maintenance and supervision of machines will increase [12]. Industrial organisations will become more team-oriented due to the integration of artificial intelligence tools, and the traditional top-down hierarchal structures will lose strength. Teamwork between co-workers as well as between workers and assistant systems will be more and more important [13]. In general, the job profiles will be demanded to carry out tasks with a much broader scope. Therefore, the workers will be expected to have a wider knowledge and expertise in several subjects [12].

The main observed consequence of the mentioned technological changes is the fast-growing demand for technological skills [12,14,15]. They include basic digital skills as well as advanced technological skills, such as programming [14]. Additionally, awareness of data security and data protection will gain more importance as a result of this demand [12]. The demand for social and emotional skills (which the machines are a long way from learning) will also rapidly increase due to the adoption of the advanced technologies [12,14] As stated, due to the increasing automation and digitalisation of industrial processes, the workforce will be responsible for more complex tasks. The execution of those tasks will require numeracy, solid literacy, problem-solving, and information and communication technologies (ICT) skills as well as soft skills of autonomy, collaboration and coordination [15,16].

Demand for cognitive skills will be mainly altered from basic to higher ones; the increasing automation of machines will decrease the amount of the tasks that demand basic cognitive skills (such as basic data processing) [14]. Higher cognitive skills, such as creativity, critical thinking, teamwork, problem-solving, decision-making, lifelong learning, and so on will become very crucial [12,14].

Furthermore, skills such as managing complexity, complex information processing and abstraction for obtaining a simplified representation of the bigger picture will be demanded from the near future workforce [15]. Skills such as decision making, critical thinking and independent problem solving will be considered crucial especially in reviewed technical profiles, such as production operators and control technicians [12]. Moreover, the demand for managerial, communication and organisational skills will increase significantly [12,15].

The required physical and manual skills for the job profiles will be redefined depending on the range of automated work activities. In general, the demand for physical and manual skills will also drop, but it will still remain the largest category of workforce skills in the near future [14].

Green skills are considered key to maintain the competitive edge of the European manufacturing industry as a result of the increased focus on environmental awareness and sustainability. Therefore, in the near future in Europe, the workforce (including the food industry) will be expected to master green skills.

Overall, as a result of Industry 4.0, the near-future workforce is expected to have more accentuated social, emotional, higher cognitive and technological skills, than basic cognitive, physical and manual skills [14]. The general trend points to a greater need for technological knowledge and less administrative and technical knowledge.

In conclusion, the food industry demands new professional skills from its workforce. Therefore, there is a high demand for a continuous update of the qualifications, skills and knowledge of its workforce in order to create a highly qualified, multi-skilled labor force that can handle all the technological progress [17,18]. Through this updating process, the food industry workforce of today and tomorrow, will be able to adapt to digital transformations, changes in production processes and newly introduced working practices and patterns of which are mostly connected with computer sciences [19,20]. Addressing the current skill needs and foreseeing the future skill requirements of the sector is the first step of the continuous skills update of its workforce. In this work, we have achieved this through the generation of a sectorial database. After that, a strategy should be developed for reducing the skills gaps between the jobs and the workforce. It should involve not only attracting and developing the new talents needed, but also re-skilling current employees through well-organised training and education programs, as well as re-designing work processes [3].

The food industry has been lacking a specific roadmap that guides the sector through the industrial revolution 4.0. The sector urgently needs a strategy for establishing and meeting current and future skills requirements. Therefore, the food industry needs to develop tools for implementing new skills and competences, for which reviewing the approaches already taken by the other sectors (construction, steel, automotive, etc.) would be very beneficial. Additionally, the sector needs to identify directives for education policy-makers, so that related degrees, subject syllabuses, and continuous training programs become aligned with these skills needs. Our work is developed in order to fill this gap. In this work, we firstly introduced a strategy adapted from several European sectorial projects to meet the future skills requirements and create a highly qualified and competent workforce. After, we focused on identifying the current and near-future key skills and competencies demanded by the professional profiles (engineers, operators and managers) of the food industry. We used ESCO’s (European Skills, Competences, Qualifications and Occupations) research as the main source for identifying the current food sector-related job profiles and needed skills for each profile. Then we generated an automated database for the current skills of these profiles using Visual Basic for Applications (VBA) in the excel format. During the development of the future skills and competences, we analysed all the job profiles present at the database one by one and we selected the ones which will be transformed by digital technological developments. We benefited from respectable European references through collecting data from the European ICT Professional Role Profiles framework and several strategic sectorial and inter-sectorial European projects. After identifying the additional skills that will be needed for each profile in the near future, we added them to our database and completed the update for the near-future skills requirements for the job profiles. Ultimately, we generated an automated database for current and future skills requirements for each professional profile, which can be used as a fundamental framework by the food sector through all the midterm-future changes caused by Industry 4.0. Therefore, the target end-users of the developed database are the companies of the food sector, the training centers and universities that are responsible for designing and delivering convenient training programs to provide the aforementioned skills needs.

We believe that our work will be a key tool providing guidelines to the food industry to build a workforce meeting the future skills requirements and guiding the sector leaders as well as the academia to offer better oriented and well-developed continuous training programs.

## 2. Materials and Methods

In this chapter, first of all, we present a strategy for the food industry to meet the current and future skills requirements in order to overcome the challenges coming with Industry 4.0, make good use of digitalisation and keep up with the latest technological innovations.

The activities encompassed by this long-term skills strategy should focus on making the workforce pro-active to the implementation of new technologies that help to optimize food manufacturing and increase efficiency. In order to reduce the time required for the full digital transformation process, these activities should constantly supervise the implementation of industrial skills in training programs and develop the required tools for that implementation [20,21,22].

The industry-driven proactive strategy should involve the next steps in order to achieve a successful digital transformation in the industry:The first step is evaluating the current state of the digital transformation of the food industry [20], and analysing the key trends about the upcoming technological developments [21]. After identifying the main technological developments and the related required skills and competencies, a future scenario can be developed. Economic development related to digital transformations should be also considered [11,20]. The manufacturing processes and workforce that are affected by the digital transformation will be determined.This step involves identifying the skills and professional job profiles that will be needed in the future and determining the skills gaps that are created by the current and the foreseen technological developments [20,21]. The most crucial part of this step is to identify the skills demands of the industry in proactive ways, taking skills gaps and shortages into account [20]. An automated database of the sector-related job profiles defining the needed skills and competencies will be generated. It will enable an internationally common ground and mutual recognition for the needed skills and jobs in the food industry.As the next step, the training and curricula requirements will be determined considering the skills gaps, and then, the training programs will be created for the selected skills and job profiles [21]. New methods should be discovered to implement education content in a rapid and effective way, not only in the companies but also in the formal education and training institutions. Training programs should be upgraded and updated continuously to reach a higher quality [20]. Talent management and recruitment processes should be also included in the training process [20].Better matching between skill requirements of the food industry and skills provided by training centers will be assured [21]. New standards will be developed for the sector skills recognition [20].The next step will be finding out new methods to attract more talented people to the food industry and improve the opportunities for a more diverse talent pool, and overcome recruitment challenges [20,21,22].The final step is called monitoring. All the adjustments coming with the strategy will be monitored continuously to adopt upcoming new developments [9].

This work involves the first two steps of the strategy.

During the execution of our work, to develop the database presented herein, we used the ESCO database and the European ICT Professional Role Profiles framework (generated by the Council of European Professional Informatics Societies (CEPIS) and European Committee for Standardization (CEN)) as the main data sources. We also took several strategic sectorial and inter-sectorial European projects as a reference, in some of which we directly took part or acted as collaborators: ESSA (steel sector) [20], DRIVES (automotive sector) [21], APPRENTICESHIPQ (Procedures for Quality Apprenticeships in Educational Organisations) [22] and SMeART ((Digitalization of Small and Medium Enterprises, SMEs) [23]. Therefore, it is compulsory to give definitions of the references in order to clarify our study. ESCO is the European multilingual classification of Skills, Competences, Qualifications and Occupations. In other words, it is a dictionary that describes, identifies and classifies professional occupations, skills, and qualifications relevant for the labor market, education and training [24]. It is directly linked to the International Standard Classification of Occupations (ISCO) which is a classification of occupation groups managed by the International Labor Organization (ILO), since the information and data in ESCO are based on an original work published by the ILO under the title “International Standard Classification of Occupations”, ISCO-08. CEPIS is a non-profit organisation seeking to improve and promote a high standard among informatics professionals in recognition of the impact that informatics has on employment, business and society [25]. CEN is an association that brings together the National Standardisation Bodies of 34 European countries supporting standardisation activities in relation to a wide range of fields and sectors [26]. The European ICT Professional Role Profiles framework was generated with the help of these two organisations in order to contribute to a shared European reference language for developing, planning and managing ICT professional needs in a long-term perspective and to maturing the ICT profession overall [27].

Additionally, during the generation of the automated database for the job profiles, VBA was used as the programming language in Microsoft Word Excel file.

## 3. Results and Discussion

Our aim was to generate an automated database for skill requirements (current and near-future) for all the professional profiles related to the food sector.

We based our work on ESCO’s research using it as the main source for identifying the current sector-related job profiles and needed skills for each profile. First, we selected the professional profiles related to the food industry in the ESCO database which is in word excel format. Once all the sector-related job profiles were extracted and integrated into a new excel file, the automation process was carried out using VBA and an automated database was generated in the excel format. This database included only the current skills needs of the job profiles.

During the development of the future skills and competences, we collected data from the European ICT Professional Role Profiles framework and benefited from several strategic sectorial and inter-sectorial European projects. In order to incorporate the near-future skills requirements of the job profiles into the database, we studied all the job profiles one by one and selected the ones which will be transformed by the digital technological developments. During this process, we used the aforementioned European projects as a guide to analyse what kind of job profiles related to these sectors (steel, automotive, smart engineering etc.) have undergone modification through digital transformation. We observed that job profiles that have common roles in all of the industrial areas (food, steel, automotive, etc.), such as project manager, production manager etc. will need the same kind of modifications in their skills requirements in the future. For that purpose, we identified job profiles in the food industry with equivalent roles in other sectors that have already undergone changes, and we selected their future skills requirements. Later, we also identified future skills needs for profiles that are specific to the food industry through a detailed analysis. Then, we added the new skills and competencies to our database and updated the needed skills for the altered job profiles. Furthermore, the European ICT Professional Role Profiles framework provided us the specific job profiles of which skills have undergone changes through Industry 4.0 as well as the additional skills that will be required by these profiles in the near future. We added the mentioned future skills of the professional profiles that were already present in our database and completed the update for the near-future skills requirements for the job profiles. Figure 1 demonstrates an example tab of the generated database for the food industry. In this specific case, the ‘food grader’ professional profile is used as an example. The first four rows of the table show a hierarchical order of the occupational groups; in this case, the job profile ‘food grader’ belongs to the ‘Food and beverage tasters and graders’ group, which is a part of the bigger occupation group ‘Food processing and related trades workers’, and so on. The database also provides a weblink to ESCO’s webpage where we can find all the introduced data related to the job profile. In the table, we can also see alternative names used for the same job profile in food sector and the ISCO number of the profile, which can be interpreted as an international code of the occupation for the International Standard Classification of Occupations (ISCO). Additionally, the table shows the essential and optional knowledge and skills currently needed for the job profile, as well as the future skills that we introduced after a detailed analysis.

The skills that are currently demanded by the food sector and extracted from the ESCO database are in black, while the future skills (essential and optional) that are generated with the help of ICT Professional Role Profiles framework European Union sectoral projects and several other sources are in red. The complete version of the table (Figure 1) is provided in Appendix A.

Since this is an automated database, we can call it a smart table. For example, in the excel table of Figure 1, when a job profile is introduced in the cell related to the professional job profile (changing ‘food grader’ to another profile), all the data related to the new profile appears automatically in the table, replacing the information related to the ‘food grader’ profile. The automation of the database allows us to reach the current and future skill needs of any professional profile immediately. This makes the database a very functional tool during the development of the training programs.

Table 1, Table 2 and Table 3 present a brief view of the last version of the automated database: the professional job profiles related to the food industry, their definition, ISCO numbers and the key skills demanded by these job profiles. All the data here is provided by ESCO. Table 4 demonstrates our contribution to the database: future skills requirements demanded by the mentioned job profiles. These newly introduced skills are mainly transversal and technological ones generated by Industry 4.0 and they are highlighted in the red colour since they represent our addition to the database. We used 7 job profiles as an example. The complete version of the documents, Table 1, Table 2 and Table 3, can be found in the Appendix A respectively.

## 4. Conclusions

Having a qualified workforce that can handle the growing technology is the major key condition for the food industry to overcome the big industrial revolution challenge. It can only be achieved through addressing the current skill needs and foreseeing future skill requirements of the sector and providing the most convenient training and education programs to reduce the skills gaps between the workforce and the industry needs.

In this work, firstly, we introduced an industry-driven proactive strategy adapted from several European sectorial projects to achieve a successful digital transformation for the food industry. It can serve the sector as a guideline to meet future skills requirements.

Our work aimed to address the near-future changes in the professional skill requirements of the food industry facing Industry 4.0. In this framework, each job profile related to the food industry in the ESCO database was analysed, whether they will undergo a change due to digitalisation or not. Then, based on the obtained answer, the skills requirements of each profile are updated depending on the range of automation of their work activities. Moreover, we identified the qualifications demanded by each professional job profile benefiting from highly acceptable European sources.

We generated an automated database for skills requirements (current and near-future) for the professional profiles related to the food industry, which can be used as a fundamental framework by the sector through all the midterm-future changes caused by Industry 4.0. Finally, it is worth mentioning that the obtained results have raised the interest of the Basque Food Cluster.

For all these reasons, we believe that our work can be a sectorial and academic guideline to prepare convenient and well-developed training programs to deliver the needed skills. Applying the successful training programs created through the food industry—academia collaboration, the gap between what is expected by the industry and what is delivered by the workforce will be bridged. The food industry will win over the new talents who gain the required specific qualifications and full expertise of the sector through these specific training programs. The training of current employees in the sector will cause the food industry to keep up with the industrial changes and global competitiveness. Therefore, the workforce with updated qualifications will increase the production efficiency of the sector.

The food industry is always searching for new projects and lines of investigation, with a special focus on those concerning the sector’s digitalisation. A major part of these researches would be ideally wider versions of our work; they would include training and educational programs created to deliver the demanded industrial skill requirements which were addressed by our research. Therefore, we believe that our work can be used as a roadmap for the next generation of projects in this area.

## Figures and Tables

**Figure 1 foods-09-00492-f001:**
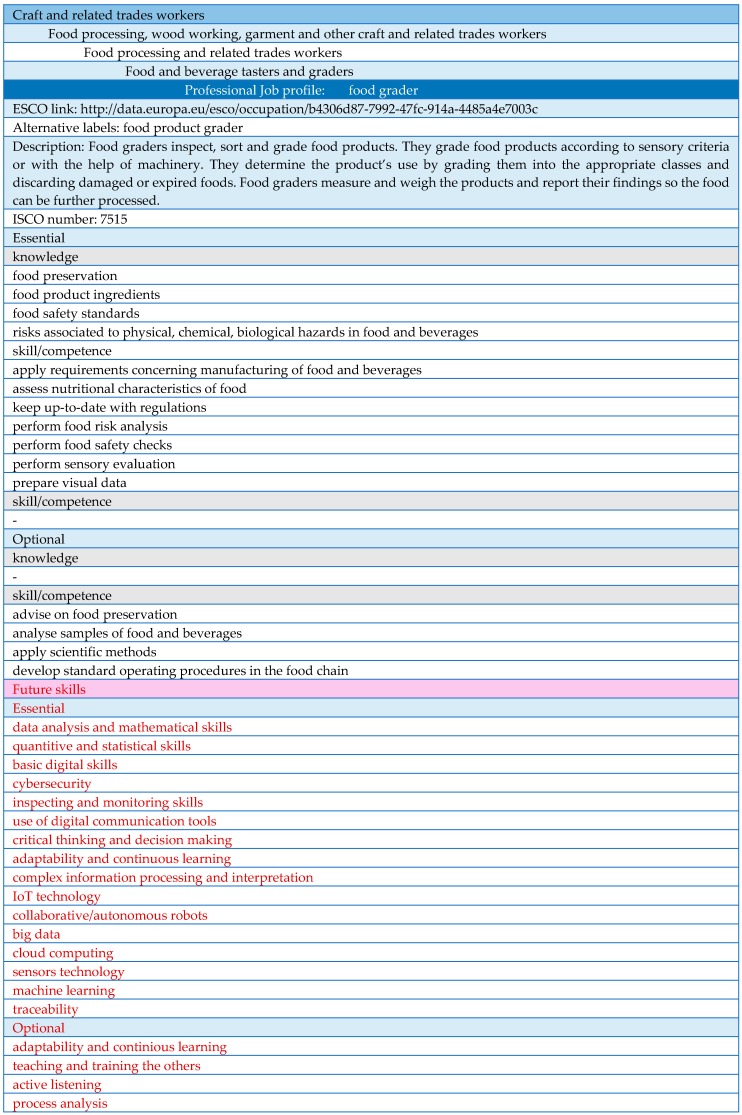
An example tab of the automated database generated for the job profiles in the food industry.

**Table 1 foods-09-00492-t001:** A brief view of the automated database (in excel format) including the description, alternative names and ISCO numbers of the professional profiles related to the food industry.

ESCO Occupation	Food Production Manager	Food Production Operator	Food Safety Specialist	Food & Beverage Packaging Technologist	Food Production Engineer	Food Analyst	Food Technician
web link to ESCO	http://data.europa.eu/esco/occupation/a7d6a377-3d3b-41cf-9301-f4a0a7ad3d96	http://data.europa.eu/esco/occupation/e3dc66de-99c7-4607-a82b-7244036d316d	http://data.europa.eu/esco/occupation/00ab5610-e715-428f-99f6-b1e5e469dbcd	http://data.europa.eu/esco/occupation/8a925aca-f636-437c-8962-5deae170e246	http://data.europa.eu/esco/occupation/2f26a52b-cf45-4282-9138-478252161f00	http://data.europa.eu/esco/occupation/33fe4c90-c4fd-4860-bdc5-24fcab16f45a	http://data.europa.eu/esco/occupation/afee6a28-f654-4ac7-a665-f758616bc689
Alternative labels	no alternative labels	food production operative//food production worker//food manufacturing worker//food worker	food production quality controller//trainee food safety specialist//food safety controller//senior food safety specialist//food scientist//food safety monitor//…	food and beverage packaging expert//food and beverage packaging specialist//food and drinks packaging technologist//food and drinks packaging expert	food engineer	food researcher//food research specialist//food analysis expert//food analysis specialist	food tech expert//food technology expert//food technology specialist//food tech specialist
Description	Food production managers oversee and monitor production and have overall responsibility for staffing and related issues. Hence, they have a detailed knowledge of the manufacturing products and their production processes. On the one hand, they control process parameters and their influence on the product and on the other hand, they ensure that staffing and recruitment levels are adequate.	Food production operators supply and perform one or more tasks in different stages of the food production process. They perform manufacturing operations and processes to foods and beverages, perform packaging, operate machines manually or automatically, follow predetermined procedures, and take food safety regulations on board.	Food safety specialists organise processes and implement procedures to avoid problems with food safety. They comply with regulations.	Food and beverage packaging technologists assess appropriate packaging for various food products. They manage matters in relation to packaging while ensuring customer specifications and company targets. They develop packaging projects as required.	Food production engineers oversee the electrical and mechanical needs of the equipment and machinery required in the process of manufacturing food or beverages. They strive to maximise plant productivity by engaging in preventive actions in reference to health and safety, good manufacturing practices (GMP), hygiene compliance, and performance of routine maintenance of machines and equipment.	Food analysts perform standardised tests to determine the chemical, physical, or microbiological features of products for human consumption.	Food technicians assist food technologists in the development of processes for manufacturing foodstuffs and related products based on chemical, physical, and biological principles. They perform research and experiments on ingredients, additives and packaging. Food technicians also check product quality to ensure compliance with legislation and regulations.
ISCO Number	1321	8160	2263	2141	2141	3111	3119

**Table 2 foods-09-00492-t002:** A brief view of the automated database (in excel format) including essential skills, knowledge and competences needed by the professional profiles related to the food industry.

Food Production Manager	Food Production Operator	Food Safety Specialist	Food & Beverage Packaging Technologist	Food Production Engineer	Food Analyst	Food Technician
**essential**	**essential**	**essential**	**essential**	**essential**	**essential**	**essential**
knowledge	knowledge	knowledge	knowledge	knowledge	knowledge	knowledge
financial capability	food safety principles	food legislation	packaging engineering	electrical engineering	food safety principles	food and beverage industry
food and beverage industry		food preservation	packaging functions	electronics	food safety standards	food preservation
food legislation		food storage	packaging processes	food storage	food science	food product ingredients
quality assurance methodologies			product package requirements	quality assurance methodologies	food toxicity	functional properties of foods
			types of packaging materials		laboratory-based sciences	processes of foods and beverages manufacturing
skill/competence	skill/competence	skill/competence	skill/competence	skill/competence	skill/competence	skill/competence
analyse production processes for improvement	administer ingredients in food production	control food safety regulations	analyse packaging requirements	apply GMP	analyse characteristics of food products at reception	apply GMP
analyse trends in the food and beverage industries	apply GMP	develop food safety programmes	apply GMP	apply HACCP	analyse samples of food and beverages	apply HACCP
apply control process statistical methods	apply HACCP	evaluate retail food inspection findings	apply HACCP	apply requirements concerning manufacturing of food and beverages	apply GMP	apply requirements concerning manufacturing of food and beverages
apply HACCP	be at ease in unsafe environments	keep task records	care for food aesthetic	configure plants for food industry	apply requirements concerning manufacturing of food and beverages	clean food and beverage machinery
apply requirements concerning manufacturing of food and beverages	carry out checks of production plant equipment	maintain personal hygiene standards	identify innovative concepts in packaging	develop food production processes	assess nutritional characteristics of food	ensure public safety and security
communicate production plan	clean food and beverage machinery	monitor packaging operations	keep up with innovations in food manufacturing	disaggregate the production plan	assess quality characteristics of food products	identify the factors causing changes in food during storage
control of expenses	disassemble equipment	plan inspections for prevention of sanitation violations	manage packaging development cycle from concept to launch	disassemble equipment	attend to detail regarding food and beverages	manage all process engineering activities
ensure cost efficiency in food manufacturing	ensure refrigeration of food in the supply chain	prepare reports on sanitation	manage packaging material	keep up with innovations in food manufacturing	blend food ingredients	manage delivery of raw materials
identify hazards in the workplace	ensure sanitation	take action on food safety violations	monitor filling machines	keep up-to-date with regulations	calibrate laboratory equipment	manage packaging material
implement short term objectives	follow production schedule	train employees	monitor packaging operations	manage all process engineering activities	collect samples for analysis	monitor freezing processes

**Table 3 foods-09-00492-t003:** A brief view of the automated database (in excel format) including optional skills, knowledge and competences needed by the professional profiles related to the food industry.

Food Production Manager	Food Production Operator	Food Safety Specialist	Food & Beverage Packaging Technologist	Food Production Engineer	Food Analyst	Food Technician
**optional**	**optional**	**optional**	**optional**	**optional**	**optional**	**optional**
knowledge	knowledge	knowledge	knowledge	knowledge	knowledge	knowledge
food safety standards	centrifugal force	cold chain	food safety principles	food and beverage industry	fermentation processes of food	cleaning of reusable packaging
legislation about animal origin products	cleaning of reusable packaging	food homogenisation	food safety standards	food homogenisation	food homogenisation	combination of flavours
	condiment manufacturing processes	food policy	food science	food preservation	food legislation	combination of textures
	fermentation processes of food	general principles of food law	ingredient threats	food safety standards	food products composition	fermentation processes of beverages
	food canning production line		risks associated to physical, chemical, biological hazards in food and beverages		risks associated to physical, chemical, biological hazards in food and beverages	fermentation processes of food
skill/competence	skill/competence	skill/competence	skill/competence	skill/competence	skill/competence	skill/competence
adapt production levels	adjust drying process to goods	analyse samples of food and beverages	assess HACCP implementation in plants	analyse work-related written reports	analyse packaging requirements	adjust production schedule
advocate for consumer matters in production plants	administer materials to tea bag machines	assess food samples	detect microorganisms	assess HACCP implementation in plants	analyse trends in the food and beverage industries	administer ingredients in food production
apply foreign language for international trade	apply different dehydration processes of fruits and vegetables	audit HACCP	develop new food products	be at ease in unsafe environments	analyse work-related written reports	analyse packaging requirements
assess environmental plans against financial costs	apply extruding techniques	develop food policy	develop standard operating procedures in the food chain	ensure compliance with environmental legislation in food production	apply scientific methods	analyse production processes for improvement
ensure continuous preparedness for audits	apply preservation treatments	ensure correct goods labelling	ensure correct goods labelling	ensure full functioning of food plant machinery	assess environmental parameters at the workplace for food products	analyse work-related written reports
hire new personnel	check bottles for packaging	monitor sugar uniformity	keep up-to-date with regulations	food plant design	assess food samples	apply control process statistical methods
lead process optimisation	check quality of products on the production line	use instruments for food measurement	label foodstuffs	lead process optimisation	assess shelf life of food products	apply food technology principles
manage medium-term objectives	conduct cleaning in place		participate in the development of new food products	write work-related reports	check quality of products on the production line	be at ease in unsafe environments
manage staff	dispose food waste			interpret extraction data	detect microorganisms	calibrate laboratory equipment

**Table 4 foods-09-00492-t004:** A brief view of the automated database (in excel format) including the future skills requirements of professional profiles related to the food industry.

Food Production Manager	Food Production Operator	Food Safety Specialist	Food & Beverage Packaging Technologist	Food Production Engineer	Food Analyst	Food Technician
future skills	future skills	future skills	future skills	future skills	future skills	future skills
**essential**	**essential**	**essential**	**essential**	**essential**	**essential**	**essential**
advanced communication and negotiation skills	advanced data analysis and mathematical skills	data analysis and mathematical skills	basic digital skills	basic digital skills	data analysis and mathematical skills	advanced data analysis and mathematical skills
leadership and managing others	use of complex digital communication tools	quantitive and statistical skills	advanced data analysis and mathematical skills	advanced data analysis and mathematical skills	quantitive and statistical skills	use of complex digital communication tools
adaptability and continuous learning	Interpersonal skills and empathy	basic digital skills	cybersecurity	advanced IT skills & Programming	basic digital skills	Interpersonal skills and empathy
critical thinking & decision making	adaptability and continuous learning	cybersecurity	use of complex digital communication tools	use of complex digital communication tools	cybersecurity	adaptability and continuous learning
personal experience	teaching and training others	inspecting and monitoring skills	advanced IT skills & programming	cybersecurity	inspecting and monitoring skills	teaching and training others
active listening	active listening	use of digital communication tools	entrepreneurship and initiative taking	advanced communication skills	use of digital communication tools	active listening
work autonomously	basic numeracy and communication	critical thinking and decision making	adaptability and continuous learning	leadership and managing others	critical thinking and decision making	basic numeracy and communication
quantitative and statistical skills	basic data input and processing	adaptability and continious learning	critical thinking & decision making	entrepreneurship and initiative taking	adaptability and continious learning	basic data input and processing
complex information processing and interpretation	advanced literacy	complex information processing and interpretation	basic numeracy and communication	adaptability and continuous learning	complex information processing and interpretation	advanced literacy
prcess analysis	complex information processing and interpretation	process analysis	basic data input and processing	teaching and training others	process analysis	complex information processing and interpretation
creativity	process analysis	initiative taking	advanced literacy	critical thinking & decision making	initiative taking	process analysis
complex problem solving	creativity	advanced IT skills & Programming	auantitative and statistical skills	active listening	advanced IT skills & Programming	creativity
basic digital skills	complex problem solving	traceability	complex information processing and interpretation	complex information processing and interpretation	traceability	complex problem solving
advanced data analysis and mathematical skills	IoT	adaptability and continuous learning	process analysis	process analysis	adaptability and continuous learning	IoT
cybersecurity	Manufacturing Execution System (MES)	IoT technology	creativity	creativity	collaborative/autonomous robots	Manufacturing Execution System (MES)
use of complex digital communication tools	traceability	cloud computing	complex problem solving	complex problem solving	IoT technology	traceability
advanced IT skills & programming	cybersecurity	big data	traceability	traceability	big data	cybersecurity
interpersonal skills and empathy	cloud computing	sensors technology	collaborative/autonomous robots	collaborative/autonomous robots	cloud computing	cloud computing
entrepreneurship and initiative taking	big data	machine learning	IoT technology	IoT technology	sensors technology	big data
Manufacturing Execution System (MES)	collaborative/autonomous robots	collaborative/autonomous robots	big data	big data	machine learning	collaborative/autonomous robots
IoT technology			cloud computing	cloud computing		
cloud computing			sensors technology	sensors technology		
big data				augmented reality		
machine learning				machine learning		
deep learning						
**optional**	**optional**	**optional**	**optional**	**optional**	**optional**	**optional**
adaptability and continuous learning	personal experience	personal experience	personal experience	interpersonal skills and empathy	personal experience	personal experience
training and teaching others	initiative taking	active listening	teaching and training others	personal experience	active listening	initiative taking
sensors technology	augmented reality	work autonomously	work autonomously	work autonomously	teaching and training the others	augmented reality
collaborative/autonomous robots	machine learning	augmented reality	active listening	quantitative and statistical skills	augmented reality	machine learning
traceability			interpersonal skills and empathy	basic data input and processing		
			machine learning			
			augmented reality

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
