# Peer review of "A Guide for the Food Industry to Meet the Future Skills Requirements Emerging with Industry 4.0"

_foods, 2020, doi:10.3390/foods9040492_

Round 1
Reviewer 1 Report
Thank you for the new release of the paper which takes into account my recommandations. In addition the paper has been a lot improved. It is a useful paper.
Reviewer 2 Report
The basic idea of research has been clarified, but the detailed purpose and scope of work are not presented in the right way. There is no appropriate description of what the authors really want to achieve and how they plan to do it. It is difficult to understand the authors' approach if without a description of the purpose and scope of work at the beginning.
I appreciate the practical work, involving the development of a database. However, I miss a specific explanation who and how will use these resources? In addition, it is difficult for me to understand how the data in the database correlate with the assumptions of Industry 4.0? Even if it was the authors' intention to refer to these requirements, in my opinion, it is not obvious.
The subject has the potential to publishing but requires rebuilding its structure: purpose, scope, methodology, practical work, tests/verification, evaluation, and conclusions. In the beginning, it is necessary to define and emphasize the purpose of building such a base and the expected benefits of its development.
I also have a few more comments:
- In chapter 1 there is a reference to literature written as [2,9,10,11]. It is best to avoid multiple references, and if it has to appear, it will be better to write it in like [2,9-11].
- In Chapter 2, the first paragraph seems unnecessary, since it follows from Chapter 1.
- Are the steps proposed in chapter 2 (1-6) are the representation of a particular methodology or are they the authors' own proposition?
- Chapter 3 present the purpose of the work, and this should be placed and highlighted at the beginning of the work.
- Images are too large, and captions should be limited to the image name only. Details on how to interpret the data presented on them should be placed in the text.
- The work requires editorial corrections (there are missing some punctuation marks).
- Improve English - the text should be checked by a native speaker.
- A reference to the literature written in the form of: "Project Title..." raises doubts. On the given www pages it is difficult to find references to the content cited in the text. It would be more appropriate to refer to some published results/reports/articles created as part of these projects.
Reviewer 3 Report
The authors present important subjects that Industry 4.0 to the table, the future demand for skills. This demand is observed not only in the food industry but also in other industries. My recommendations are:
- In chapter 1, the authors should provide examples of such "rapidly growing transformation" in the food industry;
- The authors should consider reformulate the first paragraph of chapter 2 as it presents a brief summary of the previous chapter, that is already presented;
- Figure 1 should be reformatted to get a better understanding of the skills inside of each topic. Additionally, links appear and there is reference to a "7515" not explained earlier what it is;
- Figure 2 should be reformatted to be more visually appealing. Too narrow columns that difficult reading.
Round 2
Reviewer 2 Report
The authors have answered all my earlier remarks and I am glad that they agree with my suggestions. The paper looks much better for me now.
This manuscript is a resubmission of an earlier submission. The following is a list of the peer review reports and author responses from that submission.
Round 1
Reviewer 1 Report
When I read the abstract I was very interested in the paper since I thought I will see the impact of industry 4 on food industry jobs and skills required. Yet when I read it I could not see the relationship between new professional skills requirements in the workforce of the food industry and smart technologies such as robotics, artificial intelligence (AI), internet of things (IoT) and machine learning etc.
“Quality assurance methodologies” are listed as essential knowledge. How can you assure quality without data analytics, statistics which are listed as essential future methodologies? Besides these have been required before industry 4.
I can understand traceability as a future skill since blockchain is relatively new technology.
I don’t think that this research “will be a key tool for the academy as well as the sector leaders to provide well-developed and better-oriented continuous training programs to reduce the skill mismatch between the workforce and the jobs.” Article is well written yet it does not deliver what it promises in the abstract, in my judgment.
Sorry the authors you searched different data bases identified the jobs and skills required but not jobs and skill required because of Industry 4.
Reviewer 2 Report
The aim of the paper is very interesting. The main problem is that it is almost impossible to read figure 1 and figure 2, because the font is too small. In addition, it could be nice to have a short synthesis about the results and the collected data.